# Effects of Substituting Types of Physical Activity on Body Fat Mass and Work Efficiency among Workers

**DOI:** 10.3390/ijerph18105101

**Published:** 2021-05-12

**Authors:** Jiameng Ma, Dongmei Ma, Junghoon Kim, Qiang Wang, Hyunshik Kim

**Affiliations:** 1Faculty of Sports Science, Sendai University, Miyagi 9891693, Japan; jm-ma@sendai-u.ac.jp (J.M.); dn-ma@sendai-u.ac.jp (D.M.); 2Department of Medicine and Science in Sports and Exercise, Graduate School of Medicine, Tohoku University, Sendai, Miyagi 9808574, Japan; 3Laboratory of Sports and Exercise Medicine, Korea Maritime & Ocean University, Yeongdo-Gu, Busan 49112, Korea; junghoonkim@kmou.ac.kr; 4College of Sports Science, Shenyang Normal University, Shenyang 110034, China; www24219497@synu.edu.cn

**Keywords:** worker, health promotion, obesity, labor productivity, isotemporal substitution

## Abstract

Low levels of physical activity (PA) not only increase healt h risks but also affect employee productivity. Although daily activity is interdependent with work productivity and personal health, few studies have examined how substituting physical activities would affect health and work efficiency. The present study aimed to investigate how substituting sedentary behaviors (SB) with increased PA and increasing the intensity of low-level activities during waking times affects the body fat mass and work efficiency of employees. Data were collected from 224 Japanese employees. SB, light physical activity (LPA), moderate physical activity (MPA), and vigorous physical activity (VPA) were measured using a tri-axial accelerometer, and body fat mass and work efficiency were also assessed. Analyses of the effects of substituting behaviors were based on the methods used in the isotemporal substitution model. Body fat mass decreased after substituting behaviors for 30 min per day: from SB to VPA (β = −4.800, 95% CI = −7.500; −2.100), from LPA to VPA (β = −4.680, 95% CI = −7.350; −1.980), and from MPA to VPA (β = −4.920, 95% CI = −7.680; −2.190). For work efficiency and physical activities, a higher work efficiency score was observed when substituting SB with LPA (β = 0.120, 95% CI = 0.030; 0.240), and a lower work efficiency score was observed when substituting LPA with VPA (β = −0.660, 95% CI = −1.350; −0.030). These results should help achieve greater results in promoting health and increasing work productivity by properly distributing and practicing daily physical activities during work hours.

## 1. Introduction

There is growing concern for lower labor productivity due to poor physical health among workers in Japan as risk factors for lifestyle diseases have increased due to an increase on average working-age [1,2,3]. To address this concern, the government passed regulations for the improvement of working methods as a national policy in 2019. This policy change was led by the Ministry of Health, Labor, and Welfare of Japan; the guidelines to improve and manage workers’ health at work have been implemented since 1 April 2021 after being publicly announced early in the year [4,5].

There have been demands for improvements in all working environments with goals to maintain and promote workers’ health and improve productivity in the workplace. Reports showed that a low level of physical activity (PA) and sedentary behavior (SB) for long periods of time are associated with increased risk of diseases such as diabetes, obesity, cardiovascular disease, and cancer [6]. In addition, body fat, one of the obesity indicators, is also reported to vary in its relevance depending on the intensity of physical activity [7]. Body fat percentage as an outcome is vital to evaluate the relationship between each intensity level of physical activity and sedentary behavior. An epidemiologic study on the association between workers’ level of PA and their risk of developing diseases highlighted that SB is not only physically harmful but may also impact labor productivity.

A previous study with adults from according to subjective valuation 20 countries worldwide revealed that Japan ranked first as a country with the longest sedentary time (420 min a day), whereas the average of sedentary time among 20 countries worldwide was 300 min per day [8]. In addition, a previous cross-sectional study on the association between sedentary time and productivity reported that the longer the sedentary time, the lower was the work performance, productivity, and work engagement [9]. work efficiency can lead to reduced production cost; therefore, it is an important part of a firm’s management strategy [10]. The problems related to workers’ health and labor productivity have been raised as Japan still suffers from long working hours [11]. However, there is almost no evidence about intervention studies using objective assessment. There has been a demand to accumulate studies based on scientific evidence to establish action guidelines about SB at work to improve workers’ health and productivity [1].

However, people’s behaviors are interdependent. This is because there is a limited amount of time in a day at a person’s disposal, and individuals must reduce the time spent on one activity to devote some to others. For example, to increase the time spent on moderate to vigorous exercise by 30 min, the time spent on another activity will need to be decreased by that much. This indicates that the periods of time spent on different actions in a day share an interdependent relationship. Willett et al. [12] Proposed that total energy intake should be considered when investigating the relationship between nutritional intake, especially that of major nutrients, and disease. The isocaloric replacement analysis (IS) model is defined as “a method of estimating the effect on an objective variable when one action is replaced by an equivalent amount of another action”. The IS model has been reported as an isocaloric replacement analysis model similar to the proposed analytical models—it is characterized by an interpretation that is closer to people’s real life than that offered by conventional analytical models, because it considers the interdependence of behaviors [13]. Exercise epidemiology of previous studies report that reallocating 30 min/day of sedentary time with LPA reduced mortality risk by 20% over a five-year follow-up period [14]. In addition, substituting LPA for VPA had effects on indicators of obesity such as weight and body fat mass [15]. Female adults’ weight reduced by 1.57 kg when 30 min of walking was substituted to jogging during the day; moreover, when 30 min of watching TV was substituted for walking during the day, their weight reduced by 1.14 kg [16].

Although the topics about workers’ level of PA and health problems have received public attention, the interdependence between the two are still unclear. In addition, there are limited studies that employed the IS model to investigate the association between the substitution of SB for various levels of PA and productivity. Furthermore, there has been no study that objectively assessed the activity pattern about Japanese [17]. The results of this study can be used for policy-development regarding health management at the workplace and to design specific SB reduction strategies. Moreover, they can be applied while designing impactful strategies for a population with complex demographic features.

This study aimed to identify the effects of substituting the amount of time spent in SB for other PA activities on body fat mass and productivity. We can acquire new information about the distribution of activity, health promotion, and improvement of productivity.

## 2. Materials and Methods

### 2.1. Participants

This study is a cross-sectional study, targeted workers from a large company located in M prefecture, in the northeastern region of Japan, and conducted in October 2018 as part of a health promotion project in the workplace. Among 380 workers from R company, 232 full-time workers (working over eight hours on weekdays), aged between 30 and 59, consented to and participated in the study. Written consent was obtained following the explanation of purposes, measurements, benefits, disadvantages, risks, and data publication. Any erroneous or defective data obtained from the accelerometer and InBody were excluded from the survey. A total of 224 participants (96.5%) with data points from all items that were commonly assessed were analyzed. This study was conducted with the approval of the Research Ethics Board of Sendai University (Approval Number: 27-6).

### 2.2. Measurements

#### 2.2.1. Demographic Variables

Gender, age, educational level, and employment status were investigated. Educational level was categorized into two groups: four-year university or higher, and two-year college or lower. Employment status was classified into two groups: sales and services, and office workers.

#### 2.2.2. Health Outcomes

The primary outcome was change in body weight body–mass index (BMI) and body fat mass. Anthropometric measurements were assessed by bioelectrical impedance analysis (BIA) the InBody 470 (Biospace, Tokyo, Japan) analyzer in light street clothing and without socks and shoes. BIA is a non-invasive, convenient tool that has minimal variation between observers [18]. The measurement team training held at the author’s organization.

#### 2.2.3. Sedentary Behavior and Physical Activity

A tri-axial accelerometer (Active Style Pro HJA-750C, Omron Health Care Co., Ltd. Kyoto, Japan) was used to measure sedentary time and the level of PA. Participants were instructed to wear the accelerometer around their waist, from the time they awoke until they went to bed, for 10 working days, except when bathing or swimming. This accelerometer has been validated for measuring physical activity and sedentary behavior in a controlled laboratory setting [19,20]. The device recalls synthetic acceleration at a measurement range of ± 6G and a resolution of 3m. Additionally, the device can precisely measure such as LPA, SB [19,20]. Refer to the previous studies than when the value on the accelerometer remained at 0 for 20 min or longer, it recognized that the person was not wearing the device [21]. The amount of PA was evaluated every 10 s epochs, on the following categories, using the tri-axial accelerometer: 1.5 METs or below for sedentary time, 1.6–2.9 METs for LPA, 3.0–5.9 METs for MPA, and 6 METs or above for VPA. Data for sedentary time and the amount of daily PA was extracted based on the data obtained from participants wearing the accelerometer more than 600 min per day, for four days [22]. Accelerometer data were processed using an Omron health management software, BI-LINK for physical activity professional edition Ver 2.2 and custom software (Custom-written Macro program).

#### 2.2.4. Work Efficiency

To estimate the work efficiency, some portions of the World Health Organization’s “World Mental Health Japan Survey Version” of the “WHO Health and Work Performance Questionnaire Short Form” were referenced. Previous studies report that WHO was reliability 0.95, and can be analyzed either as one factor or by extracting the unique variance for each subdimension [23]. The questions posed included, “On a scale from 0 to 10, where 0 represents the worst job performance any person could execute at your job, and 10 represents the performance of a top worker, how would you rate the typical performance of most workers in a job similar to yours?”, “Using the same 0 to 10 scale, how would you rate your typical job performance over the past year or two?”, “Using the same 0 to 10 scale, how would you rate your overall job performance on your working days in the past four weeks (28 days)?”. The associations between work-related sedentary behavior and current productivity compared with past own productivity in the present study indicate that strategies for reducing work-related sedentary behavior are needed to improve productivity [24]. The average scores for the answers to the above three questions were calculated to determine productivity [24].

### 2.3. Statistical Analysis

Furthermore, the ratio, average, and standard deviation of demographic basic variables, health outcomes, level of each PA obtained from the accelerometer, and the items evaluating the work efficiency were calculated. To evaluate the effects of substituting one type of PA for another, body fat mass and work efficiency scores (continuous variable) were identified as dependent variables, and SB, LPA, MPA, and VPA were identified as independent variables. Linear regression analysis was used to test the single factor, partition, and IS models. To facilitate the analysis of the results, the regression coefficient for each PA was changed to 1 unit per 30 min.

In addition, sex, age, and employment status, in association with SB and PA, were identified as covariates based on the results of a previous study on workers [25] and were also applied in this study. In the single factor model, the scores of body fat mass and work efficiency were dependent variables, and wear time, sedentary time, and LPA, MPA, and VPA were independent variables. In addition, sex, age, educational level, and employment status were included as covariates, and the overall association for each PA was evaluated.

In the partition model, the sedentary time, all times spent during LPA, MPA, and VPA, and covariates were applied, and the independent variable of each activity was evaluated without applying the wear time. In the IS model, sedentary time and the time spent during LPA, MPA, and VPA, except for one PA, were applied to the linear regression model as well as the wear time of the accelerometer and the covariates. For example, LPA, MPA, and VPA, the wear time, and the covariates, including sex, age, and employment status, were used to evaluate the effects of substituting SB for LPA on productivity. A statistical significance level of less than 5% was used for all factors, and SPSS ver. 25 was used for the analyses (IBM, Armonk, NY, USA).

## 3. Results

Of the 224 participants in this study, 180 were male (80%) (Table 1). The average age of the participants was 44.3 ± 9.9. Regarding educational level, 63.4% graduated from four-year universities, and 24.6% were office workers in terms of employment status. Considering health outcomes, the average body fat mass measured using InBody was 23.4 ± 6.0%, the average weight was 67.3 ± 11.9 kg, and the average BMI was 23.1 ± 3.1 kg. The average number of hours spent for each PA during a working day measured by the accelerometer was 408.2 ± 78.3 min for SB, 22.6 ± 63.9 min for LPA, 82.4 ± 21.2 min for MPA, and 4.7 ± 8.5 min for VPA. In addition, the average wear time of the accelerometer was 720.1 ± 100.9 min and the average score of work efficiency was 6.0 ± 1.4.

Regarding the relationship between body fat mass and each PA, including LPA, MPA, and VPA, in the single model, only VPA clearly had a significant relationship with body fat mass, as shown in Table 2 (β = −4.560, 95% CI = −7.200; −1.920, *p* = 0.001). In the partition model, there was a statistically significant negative correlation between LPA and body fat mass (β = −0.450, 95% CI = −0.810; −0.090, *p* = 0.012) and VPA and body fat mass (β = −5.010, 95% CI = −7.710; −2.310, *p* ≤ 0.001). The following addresses the changes in body fat mass according to the substitution of each type of PA for the others. In the IS model, the body fat mass was low when 30 min per day of SB were substituted for VPA (β = −4.800, 95% CI = −7.530; −2.010, *p* = 0.001), 30 min per day of LPA were substituted for VPA (β = −0.660, 95% CI = −1.350; −0.030, *p* = 0.048), and 30 min per day of MPA were substituted for VPA (β = −4.920, 95% CI = −7.680; −2.190, *p* ≤ 0.001) (Table 2).

For the relationship between work efficiency and each PA, in the single model, the duration of LPA was significantly positively correlated with the scores of work efficiency (β = 0.150, 95% CI = 0.030; 0.240, *p* = 0.010). In the partition model, the duration of LPA was also significantly positively correlated with work efficiency scores (β = 0.150, 95% CI = 0.060; 0.240, *p* ≤ 0.001). For the changes in work efficiency due to the substitution of one type of PA for others, in the IS model, when 30 min a day of SB were substituted for LPA, the work efficiency was high (β = 0.120, 95% CI = −0.030, 0.240, *p* = 0.015); when 30 min a day of LPA were substituted for VPA, the work efficiency was low (β = −0.660, 95% CI = −1.350; −0.030, *p* = 0.048); and when 30 min a day of VPA were substituted for LPA, the work efficiency was the highest (β = 0.240, 95% CI = 0.060; 0.420, *p* = 0.016) (Table 3).

## 4. Discussion

This study evaluated the relationship between body fat mass and the substitution of one type of PA for others, during the waking times of the day, using a tri-axial accelerometer. Substituting VPA for SB, LPA, and MPA resulted in a reduction of body fat mass; moreover, when the amount of time spent on SB was replaced with VPA, the effect was considerably more significant. In the field of public health, individuals’ inactive SB negatively affects their health status, thus emphasizing the need to promote a reduction in SB.

One study using the IS model, reported that the risk of developing metabolic syndromes decreased for individuals aged between 50 and 64 when the time spent for SB was replaced with other PA, regardless of its intensity [26]. In addition, an interventional study clearly stated that the risk of developing lifestyle diseases can be reduced by increasing the time spent for LPA; thus, decreasing the sedentary time [27]. However, there was no correlation between SB and body fat mass in any of the models used in this study. In the single model that did not make any correction between each PA, only VPA was associated with body fat mass. The partition model that made a correction between each PA, LPA, and VPA were in a statistically significant, correlated relationship with body fat mass [7]. These results may be because people with high LPA and VPA typically present a low body fat mass. In the IS model that evaluated the effects of substituting one type of activity for others, replacing SB with LPA and MPA did not have any effect on body fat mass reduction as a result of applying and adjusting each epidemiological factor as dependent variables. Therefore, the reduction in body fat mass cannot be achieved unless SB is substituted for an equivalent amount of VPA.

Studies showed that the factors promoting obesity were related not only to exercise but also to the overall lifestyle, including elements such as diet [28] and sleep [29]. Reducing sedentary time without increasing intense activity has less impact on obesity and body fat mass. The type of PA performed during the time reallocated by reducing sedentary time is considered most important. Like SB, a correlation between the reduction in body fat mass and substitution of LPA or MPA for VPA was discovered. According to the exercise guidelines of the American Sports Medical Association, increasing the amount of activity is important for reducing obesity [30]. Therefore, given the same amount of time to perform PA, performing VPA is considered to be more effective in losing weight than performing LPA or MPA. As a result of technological innovation, mechanization and automation are progressing, and the work environment is transforming into one where less energy is consumed. To prevent and manage obesity in such work environments, white-collar workers who work in a sitting position must perform PA identical to VPA during their free time.

The results of evaluating the relationship between work efficiency and each PA identified a relationship between LPA and work efficiency in both the single and partition models, i.e., the more LPA, the higher the work efficiency was. In the IS model, when SB was substituted for LPA, there was an improvement in work efficiency. A cross-sectional study that targeted Japanese workers revealed that extended sedentary time at work was associated with low work efficiency among workers in their 20 s and 30 s and low work efficiency among workers in their 40 s and 50 s [31]. It was noted that changing work environments by introducing a standing desk was effective in reducing sedentary time, improving work engagement, and quality of life [32].

Thus, substituting SB for LPA is an effective strategy for improving work efficiency among workers. Modifying behavior from SB to LPA is easier to facilitate compared to other PAs with varying levels of intensity. Working from a standing position, taking a walk, and lightly stretching every 30 min to avoid SB for long periods of time are not only expected to improve employees’ work efficiency but also their labor productivity. When LPA was substituted for VPA, work efficiency reduced. A report on exercise for workers, especially white-collar workers, revealed that the more frequently SB and PA were performed independently, the more the work efficiency increased [33]. However, most of the PAs performed were LPA, (such as stretching) and the report did not state the effect of performing VPA on work efficiency. Domestic and international guidelines for PA stated that MPA and VPA are beneficial for disease prevention and health promotion [34,35]. For work efficiency, work-related VPA had a significant relationship with poor mental health [36], and VPA following the workday is the cause of fatigue and insomnia [37]. Thus, workers were unable to recover the following day, which negatively affected their work efficiency. Therefore, excessive PA can potentially reduce work efficiency, and adequate LPA may improve work efficiency.

This study exhibited several limitations. First, it should be noted that the IS model simply represents the changes in the dependent variables when behavioral variables change. This study is a cross-sectional study; it can only estimate the effects of changes in the dependent variables and cannot alter the variables. Hence, an investigation of the effects of long-term behavioral changes on body fat mass and work efficiency through a longitudinal and large-scale cohort study is imperative in the future. Dependent upon the workers’ type of PA and duration, occupations that affect work efficiency may differ. In future studies, it is necessary to consider the compatibility between work-related PA and PA during leisure time in addition to the effects of changing the intensity of PA. Furthermore, additional information is required for verification of the effects of the interdependence of SB, LPA, MPA, and VPA on health problems and work efficiency while considering sleep time as sleep is included in people’s activities during the day. In addition, in the algorithm development study using the tri-axial accelerometer, TAU/TAF ratio was higher more than zero in all participants, despite being 0 in non-wear time. Therefore, we believe that the probability of misclassification between non-wear time and sedentary time is very small [21].

Although this study has limitations, no other study has investigated each PA, obesity, and work efficiency using the IS model as far as we are aware. This study is extremely valuable, as the originality of this study and it evaluated the effect of substituting one type of PA for others during waking time on work efficiency. The results of this study can be used to properly allocate PAs during the day at a scheduled time to achieve the goals of health promotion and labor productivity improvement, which could also help advance exercise epidemiology and clinical sites.

## 5. Conclusions

The present study was based on the methods used in the isotemporal substitution model, aiming to investigate how substituting SB with increased PA and increasing the intensity of low-level activities during waking times affects the body fat mass and work efficiency of employees. The results showed that body fat mass decreased after substituting behaviors for 30 min per day: from SB, LPA, and MPA to VPA. For work efficiency and physical activities, a higher work efficiency score was observed when substituting SB with LPA, and a lower work efficiency score was observed when substituting LPA with VPA. These findings suggest that the appropriate allocation and practice of daily physical activity during working hours can have significant results in improving health and work productivity.

## Figures and Tables

**Table 1 ijerph-18-05101-t001:** Characteristics of the participants (*N* = 224).

Category	N	%	Mean	SD
**Characteristics**				
Sex; men	180	80.4	-	-
Age; years	-	-	44.3	9.9
**Educational level**	-	-		
4 years of university or higher	142	63.4	-	-
2 years of college or less	82	36.6	-	-
**Employment status**				
Office clerk	55	24.6	-	-
Sales, services	169	75.4	-	-
**Health outcomes**				
body fat mass (%)	-	-	23.4	6.0
Weight (kg)	-	-	67.3	11.9
BMI (kg/m^2^)	-	-	23.1	3.1
**Physical activity outcomes**				
Sedentary time (min/workday)	-	-	408.2	78.3
LPA ^a^ (min/workday)	-	-	22.6	63.9
MPA ^b^ (min/workday)	-	-	82.4	21.2
VPA ^c^ (min/workday)	-	-	4.7	8.5
Wear time (min/workday)	-	-	720.1	100.9
Work efficiency (scores)	-	-	6.0	1.4

^a^ Light physical activity; ^b^ Moderate physical activity; ^c^ Vigorous physical activity.

**Table 2 ijerph-18-05101-t002:** Single, partition, and isotemporal substitution of each physical activity per 30-min/day increase and body fat mass change.

	SB, *β* (95% CI)	*p*-Value	LPA, *β* (95% CI)	*p*-Value	MPA, *β* (95% CI)	*p*-Value	VPA, *β* (95% CI)	*p*-Value
**Single**	0.120 (−0.300; 0.540)	0.552	−0.012 (−0.570; 0.330)	0.574	−0.090 (−1.230; 1.020)	0.861	−4.560 (−7.200; −1.920)	0.001
**Partition**	−0.300 (−0.600; 0.030)	0.078	−0.450 (−0.810; −0.090)	0.012	0.030 (−1.080; 1.110)	0.978	−5.010 (−7.710; −2.310)	<0.001
**IS**								
Replace SB	Dropped		−0.210 (−0.630; 0.240)	0.356	0.270 (−0.084; 1.380)	0.629	−4.800 (−7.500; −2.100)	0.001
Replace LPA	0.120 (−0.330; 0.540)	0.622	Dropped		0.360 (−0.810; 1.530)	0.546	−4.680 (−7.350; −1.980)	0.001
Replace MPA	−0.240 (−0.900; 0.450)	0.498	−0.390 (−11.400; 0.330)	0.280	Dropped		−4.920 (−7.680; −2.190)	<0.001
Replace VPA	0.060 (−0.720; 0.870)	0.874	−0.060 (−0.870; 0.750)	0.874	−0.030 (−1.380; 1.290)	0.958	Dropped	

Adjusted for sex, age, education status, employment status.

**Table 3 ijerph-18-05101-t003:** Single, partition, and isotemporal substitution of each physical activity per 30-ninute/day increase and work efficiency (scores) change.

	SB, *β* (95% CI)	*p*-Value	LPA, *β* (95% CI)	*p*-Value	MPA, *β* (95% CI)	*p*-Value	VPA, *β* (95% CI)	*p*-Value
**Single**	−0.090 (−0.180; 0.030)	0.118	0.150 (0.030; 0.180)	0.010	0.060 (−0.210; 0.330)	0.621	−0.600 (−1.260; 0.060)	0.073
**Partition**	0.030 (−0.030; 0.120)	0.330	0.150 (0.060; 0.240)	<0.001	0.150 (−0.012; 0.420)	0.301	−0.570 (−1.230; 0.090)	0.095
**IS**								
Replace SB	Dropped		0.120 (0.030; 0.210)	0.015	0.120 (−0.150; 0.390)	0.413	−0.570 (−1.230; 0.090)	0.084
Replace LPA	−0.090 (−0.180; 0.090)	0.120	Dropped		0.030 (−0.240; 0.330)	0.765	−0.660 (−1.350; −0.030)	0.048
Replace MPA	0.030 (−0.150; 0.180)	0.743	0.150 (−0.030; 0.330)	0.084	Dropped		−0.510 (−1.170; 0.180)	0.143
Replace VPA	0.120 (−0.060; 0.300)	0.232	0.240 (0.060; 0.420)	0.016	0.180 (−0.150; 0.480)	0.285	Dropped	

Adjusted for sex, age, education status, employment status.

## Data Availability

Data provided in this study are available upon request by the corre-sponding author.

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
