# Peer review of "Effects of Substituting Types of Physical Activity on Body Fat Mass and Work Efficiency among Workers"

_ijerph, 2021, doi:10.3390/ijerph18105101_

Round 1

Reviewer 1 Report

1) I am reviewing this manuscript on March 30, 2021, but the first paragraph says the policy and the guidelines to improve and manage workers’ health at work were implemented since April 1, 2021.

2) Please explain the isotemporal substitution model in detail. It is hard to follow the analysis results without understanding the model. The following article could be a good reference.

Mekary, R. A., Willett, W. C., Hu, F. B., & Ding, E. L. (2009). Isotemporal substitution paradigm for physical activity epidemiology and weight change. American journal of epidemiology170(4), 519-527.

3) Research design of this study is very unclear. It is also very unclear about the study procedure. The authors claim that they utilized the data from the 2018 longitudinal study without providing any detail about the study (e.g., how many time points). As a result, there are too many unknowns about the data. For example, it is unclear if the authors used longitudinal data or not, when the main outcomes and physical activity measures were assessed and/or how many times. It is even unclear if this study is an intervention study or using observational data collected longitudinally or anything else. It became clearer as I personally search for some details about the isotemporal substitution model, which indicates how little information was provided for the statistical method used to analyze the data and even the data itself. FYI, the statistical model used for analysis needs to be specified as an isotemporal substitution model. The model can be fit by using a linear regression model. The following articles might be a good reference to make the manuscript better in future submissions.

Wei, J., Xie, L., Song, S., Wang, T., & Li, C. (2019). Isotemporal substitution modeling on sedentary behaviors and physical activity with depressive symptoms among older adults in the US: The national health and nutrition examination survey, 2007–2016. Journal of affective disorders257, 257-262.

4) It is not clear why some portions, not 100%, of the WHO form, was used to measure work efficiency. Please provide the reasoning behind it.

5) The work efficiency includes the evaluation of the work performance of not only self but also colleagues, over the past year or two. Two issues here. First, if this variable, work efficiency, is measuring the evaluation of the work performance of others, then it is unclear how this can be associated with one’s PA or SB. Second, if this variable is measuring any sort of evaluation in the past two years, then it is unclear how this can be associated with PA and SB at the time of the survey. Further discussion and detail are absolutely necessary.

6) Please add min and max to Table 1.

7) Please check typos in all tables. For example, there is “0” before the negative sign in the cell for the 95% CI of VPA for Single in Table 2.

8) The writing is poorly organized. Some key and necessary information can be found throughout the manuscript, but it is scattered all over the manuscript.

Author Response

Reviewer 1:

  1. I am reviewing this manuscript on March 30, 2021, but the first paragraph says the policy and the guidelines to improve and manage workers’ health at work were implemented since April 1, 2021.

Thank you very much for your comment. The author intended to convey “This would be implemented from April 1” at the time of submission, but the sentence in English was expressed inappropriately. The author would like to retain the same sentence as April 1 has passed.

  1. Please explain the isotemporal substitution model in detail. It is hard to follow the analysis results without understanding the model. The following article could be a good reference.

Mekary, R. A., Willett, W. C., Hu, F. B., & Ding, E. L. (2009). Isotemporal substitution paradigm for physical activity epidemiology and weight change. American journal of epidemiology, 170(4), 519-527.

Thank you very much for your comment. As per the reviewer’s suggestion, the author has added the concept of the IS model. (Please see, lines 63-82 of our revised manuscript).

  1. Research design of this study is very unclear. It is also very unclear about the study procedure. The authors claim that they utilized the data from the 2018 longitudinal study without providing any detail about the study (e.g., how many time points). As a result, there are too many unknowns about the data. For example, it is unclear if the authors used longitudinal data or not, when the main outcomes and physical activity measures were assessed and/or how many times. It is even unclear if this study is an intervention study or using observational data collected longitudinally or anything else. It became clearer as I personally search for some details about the isotemporal substitution model, which indicates how little information was provided for the statistical method used to analyze the data and even the data itself. FYI, the statistical model used for analysis needs to be specified as an isotemporal substitution model. The model can be fit by using a linear regression model. The following articles might be a good reference to make the manuscript better in future submissions.

Wei, J., Xie, L., Song, S., Wang, T., & Li, C. (2019). Isotemporal substitution modeling on sedentary behaviors and physical activity with depressive symptoms among older adults in the US: The national health and nutrition examination survey, 2007–2016. Journal of affective disorders, 257, 257-262.

Thank you very much for valuable question. This was a cross-sectional study, not a longitudinal one. The author has revised the statements that could have been misinterpreted. Although there were comments on linear regression analysis as the analytical method, this study used linear regression analysis for the IS model. (Please see, lines 159-160 of our revised manuscript).

  1. It is not clear why some portions, not 100%, of the WHO form, was used to measure work efficiency. Please provide the reasoning behind it.

Thank you very much for your comment. Regarding the WHO questionnaire, as pointed out by the reviewer, the following was done: (1) the author extracted one of the questionnaire items, (2) has provided the analytical information, and (3) has added supporting evidence from the literature (citations). (Please see, lines 140-142 of our revised manuscript).

  1. The work efficiency includes the evaluation of the work performance of not only self but also colleagues, over the past year or two. Two issues here. First, if this variable, work efficiency, is measuring the evaluation of the work performance of others, then it is unclear how this can be associated with one’s PA or SB. Second, if this variable is measuring any sort of evaluation in the past two years, then it is unclear how this can be associated with PA and SB at the time of the survey. Further discussion and detail are absolutely necessary.

Thank you very much for your comment. As the reviewer pointed out, evaluating current and past work is necessary because it is associated with work-related activities and helpful in developing strategies to reduce sedentary behavior. Additional information, including supporting evidence from the literature (citations), has been incorporated. (Please see, lines 148-151 of our revised manuscript).

  1. Please add min and max to Table 1.

Thank you very much for your comment. The author reviewed Table 1 but could not find what the reviewer had pointed out. We will revise it after the reviewer once again identifies the portion requiring improvement.

  1. Please check typos in all tables. For example, there is “0” before the negative sign in the cell for the 95% CI of VPA for Single in Table 2.

Thank you very much for your comment. As per the reviewer’s suggestion, Table 2 has been revised, and the entire sentence reviewed and edited. (Please see, Table 2 of our revised manuscript).

  1. The writing is poorly organized. Some key and necessary information can be found throughout the manuscript, but it is scattered all over the manuscript.

Thank you very much for your comment. As per the reviewer’s suggestion, the author reviewed the entire paper while carefully reading the reviewer’s comments, and revised the portions highlighted to ensure appropriate and clear expressions.

Reviewer 2 Report

This is an interesting paper which addresses the important topic of PA and SB in working adults. I have provided below a number of specific comments which I feel should be addressed before the paper can be considered further.

Line 38. Is this the correct date ‘April 1, 2021’ given this hadn’t happened when the paper was submitted?

Lines 40-46. I feel a stronger rationale needs to be added to the introduction for why the paper chose to look at body fatness and work efficiency. Regarding adiposity, a brief rational could be added in this section of the paper where the authors mention chronic diseases. We know that overweight and obesity is a major risk factor for chronic diseases, and chronic diseases are the number 1 cause of premature mortality globally. Linking weight gain to chronic disease risk here would help strengthen the rationale for why this study specifically looked at body fatness as an outcome. Later in the introduction, a stronger rationale for looking at work efficiency could also be added, by discussing briefly for example why this is important to organisations/countries, and what are the associated costs of reduced work efficiency. It would also be good to include why these two outcomes are being studied in this paper, is there evidence that the 2 are related?

Line 46. Add a reference at the end of this sentence to provide a source for this statement.

Lines 47-49. As a note of caution, the data reported in reference 7 have relied upon self-reported sitting times using a single-item question, which we know can vastly underestimate sedentary time compared to domain-specific measures and device-based measures (e.g. http://www.ncbi.nlm.nih.gov/pubmed/21946087). It may be worth highlighting this issue here, it can still be acknowledged that high sitting times were seen in Japan, however these are probably underestimates of the problem. National surveillance using device-based measures have revealed much higher sitting times than those seen in reference 7 (e.g. https://www.ncbi.nlm.nih.gov/pmc/articles/PMC3527832/)

Line 59. I don’t feel the following text on this line is needed, this could be removed ‘from the perspective of sports biomechanics.’

Lines 63-66. I don’t feel the text related to children’s long jump records is relevant to this paper, I’d restrict the examples given here to adult data, and preferably to studies using working adults.

Lines 67-68. Is the text provided here written the wrong way around? ‘Female adults' weight reduced by 1.57kg when 30 minutes of jogging was substituted for walking during the day’, should this be ‘Female adults' weight reduced by 1.57kg when 30 minutes of walking was substituted for jogging during the day’?

Line 73-74. The following statement is a bit unclear ‘Further, there has been no study that objectively assessed the activity pattern [13].’ Do the authors mean in Japanese samples here, as I am pretty sure studies have looked at aspects of this in other working populations using device based measures?

Lines 73-77. It may be a bit premature here to say how the study findings could be used, this sentence may be better off in the discussion.

Line 103. I recommended adding some further information on the InBody machine, for example, how does this machine measure body composition? Is there validity and reliability evidence to support its use?

Lines 111-112. The text here should be re-worded slightly ‘Additionally, the device can precisely measure LPA, such as SB’. Do you mean ‘the device can precisely measure LPA and SB’, given LPA and SB are very different behaviours?

Lines 107-118. This section on the accelerometer could be expanded to provide further details on this measurement tool. For example, how exactly does the device categorise times spent in each PA behaviour (LPA, MPA, VPA), and SB time? What software was used to initialise, download and process the data from this device? What is the validity evidence behind this device?

Lines 112-114. I am a little concerned about this statement ‘When the value on the accelerometer remained at 0 for 20 min or longer, it recognized that the person was not wearing the device’. If there was no movement for 20 minutes, could this not represent sitting still? How confident are the authors that sedentary behaviour hasn’t been misclassified as non-wear? Usually in adult data, continuous strings of zero counts from an accelerometer lasting 60 minutes or longer is classified as non-wear as it is recognised that adults can sit stationary for prolonged periods.

Line 131. Were all variables normally distributed, which justifies why the mean and SD are used as descriptives?

Line 159. As body composition is naturally different between men and women, I recommend reporting the body fat percentage data separately for men and women.

Table 1. This table repeats all of the information reported in the paragraph above, I’d either remove the table or remove all data in the first paragraph of the results section to ensure there is no repetition between the text and table.

Table 1. Within this table, are the accelerometer data referring to work time only? Or total waking time on workdays? I recommend making it very clear in the methods how the accelerometer data were processed and what specific data were used in the analysis. If only work time data are used, the methods need to include details on how work times were known. Did participants provide a daily diary where they reported their start and finish work times for example? If work time data are used only, what is the rationale for this? It could be worth exploring also whether similar associations exist between leisure time behaviours and body fatness and work efficiency. Presumably, if participants wore the device for 10 days, both work days and non-workdays would have been captured?

Line 195. Linked to the above point, at the start of the discussion, the text states here ‘This study evaluated the relationship between body fat mass and the substitution of one type of PA for others, during the waking times of the day, using…’. Should this be working times? The reason for the confusion is in Table 1, if you add up the time spent in each behaviour (SB, LPA, MPA, VPA) you get 518 minutes, which is much lower than the total daily wear time of 720 minutes, making me think the analyses are based on work time only?

Discussion, general comments. From looking at the PA data of the sample, they appear to be quite active, e.g. accumulating 82 mins/day of MPA. It would be worth discussing the context of the sample in more detail, in terms of the types of activities and work they engaged in. It appears that the sample weren’t typical office-based workers? The discussion could be strengthened by discussing the high activity levels of this sample, and whether these findings are relevant to more sedentary office-based workers. It would also be important to discuss briefly whether the characteristics of those taking part in the study differ in any way to those who chose not to participate. For example, are the participants used in the analysis representative of all of those employed in their organisation? The discussion also mentions the impact of sit-stand desks for office workers, more recent RCTs have been conducted in this area which show favourable outcomes on job engagement, which could be useful to refer to (e.g. https://www.bmj.com/content/363/bmj.k3870)

Author Response

This is an interesting paper which addresses the important topic of PA and SB in working adults. I have provided below a number of specific comments which I feel should be addressed before the paper can be considered further.

  1. Line 38. Is this the correct date ‘April 1, 2021’ given this hadn’t happened when the paper was submitted?

Thank you very much for your comment. The author intended to convey “This would be implemented from April 1” at the time of submission, but the sentence in English was expressed inappropriately. The author would like to retain the same sentence as April 1 has passed.

  1. Lines 40-46. I feel a stronger rationale needs to be added to the introduction for why the paper chose to look at body fatness and work efficiency. Regarding adiposity, a brief rational could be added in this section of the paper where the authors mention chronic diseases. We know that overweight and obesity is a major risk factor for chronic diseases, and chronic diseases are the number 1 cause of premature mortality globally. Linking weight gain to chronic disease risk here would help strengthen the rationale for why this study specifically looked at body fatness as an outcome. Later in the introduction, a stronger rationale for looking at work efficiency could also be added, by discussing briefly for example why this is important to organisations/countries, and what are the associated costs of reduced work efficiency. It would also be good to include why these two outcomes are being studied in this paper, is there evidence that the 2 are related?

Thank you very much for your valuable question. As the reviewer stated, there have been many studies on physical activity and sedentary behavior and their relationship with obesity, as well as on identifying weight, physical activity, and sedentary behavior as obesity indicators. Among these are studies on the relationship between the different intensity levels of physical activity and obesity. Obesity is an important indicator of health, and several studies have reported that an extremely high body fat percentage increases the risk of chronic diseases. Therefore, body fat percentage was selected as an outcome in this study. In addition, there have been many reports on the relationship between work efficiency and physical activity, and recently, on sedentary behavior in particular. Regarding work efficiency and cost, previous studies have reported a negative correlation because as work efficiency increases, the cost decreases. In this regard, this study selected body fat percentage and work efficiency as the outcomes. The author has added citations that provide supporting evidence for the same as follows. (Please see, lines 44-47, 54-62 of our revised manuscript).

  1. Add a reference at the end of this sentence to provide a source for this statement.

Thank you very much for your comment. As per the reviewer’s suggestion, references have been added. (Please see, reference 4 of our revised manuscript).

  1. Lines 47-49. As a note of caution, the data reported in reference 7 have relied upon self-reported sitting times using a single-item question, which we know can vastly underestimate sedentary time compared to domain-specific measures and device-based measures (e.g. http://www.ncbi.nlm.nih.gov/pubmed/21946087). It may be worth highlighting this issue here, it can still be acknowledged that high sitting times were seen in Japan, however these are probably underestimates of the problem. National surveillance using device-based measures have revealed much higher sitting times than those seen in reference 7 (e.g. https://www.ncbi.nlm.nih.gov/pmc/articles/PMC3527832/)

Aikaterini Grimani 1 2, Emmanuel Aboagye 1, Lydia Kwak. The effectiveness of workplace nutrition and physical activity interventions in improving productivity, work performance and workability: a systematic review. BMC Public Health. 2019 Dec 12;19(1):1676. doi: 10.1186/s12889-019-8033-1.

Thank you very much for your comment. As the reviewer pointed out, the study could have overestimated the results because it employed the self-report method by asking respondents to answer only one item from IPAQ. However, the results of studies employing the same research method in other countries have shown the sedentary behavior time in Japan to be the highest; therefore, the author believes there is a correlation. In addition, the author has highlighted the presence of bias and added the phrase “based on subjective investigation.” (Please see, lines 51-52 of our revised manuscript).

  1. Line 59. I don’t feel the following text on this line is needed, this could be removed ‘from the perspective of sports biomechanics.’

Thank you very much for your comment. As per the reviewer’s suggestion, the sentence was revised to convey the point appropriately. (Please see, lines 50-51 of our revised manuscript).

  1. Lines 63-66. I don’t feel the text related to children’s long jump records is relevant to this paper, I’d restrict the examples given here to adult data, and preferably to studies using working adults.

Thank you very much for your comment. As per the reviewer’s suggestion, the author has removed the mention of studies on children, and added citations of studies on physical activity and health status among adults. (Please see, lines 76-78 of our revised manuscript).

  1. Lines 67-68. Is the text provided here written the wrong way around? ‘Female adults' weight reduced by 1.57kg when 30 minutes of jogging was substituted for walking during the day’, should this be ‘Female adults' weight reduced by 1.57kg when 30 minutes of walking was substituted for jogging during the day’?

Thank you very much for your question. As pointed out by the reviewer, we have modified the sentence readers to understand. (Please see, lines 79-80 of our revised manuscript).

  1. Line 73-74. The following statement is a bit unclear ‘Further, there has been no study that objectively assessed the activity pattern [13].’ Do the authors mean in Japanese samples here, as I am pretty sure studies have looked at aspects of this in other working populations using device based measures?

Thank you very much for your question. The explanation was insufficient. The author has added “about Japanese” as the study focused on Japan. (Please see, lines 87 of our revised manuscript).

  1. Lines 73-77. It may be a bit premature here to say how the study findings could be used, this sentence may be better off in the discussion.

Thank you very much for your comment. The author believes significant revision of the relevant portions, as per the reviewer’s comments, is necessary as it helps highlight the need for this study and is related to the research purpose.

  1. Line 103. I recommended adding some further information on the InBody machine, for example, how does this machine measure body composition? Is there validity and reliability evidence to support its use?

Thank you very much for your question. (Please see, lines 114-118 of our revised manuscript).

  1. Lines 107-118. This section on the accelerometer could be expanded to provide further details on this measurement tool. For example, how exactly does the device categorise times spent in each PA behaviour (LPA, MPA, VPA), and SB time? What software was used to initialise, download and process the data from this device? What is the validity evidence behind this device?

Thank you very much for your question. Detailed information about the accelerometer has been added and significant revisions made. (Please see, lines 131-136 of our revised manuscript).

  1. Lines 112-114. I am a little concerned about this statement ‘When the value on the accelerometer remained at 0 for 20 min or longer, it recognized that the person was not wearing the device’. If there was no movement for 20 minutes, could this not represent sitting still? How confident are the authors that sedentary behaviour hasn’t been misclassified as non-wear? Usually in adult data, continuous strings of zero counts from an accelerometer lasting 60 minutes or longer is classified as non-wear as it is recognised that adults can sit stationary for prolonged periods.

Thank you very much for your comment. According to previous studies that evaluated the use of the same accelerometer on the same participants, inactivity over 20 minutes was treated as the participant not wearing the device. This information has been added in the paper. (Please see, lines 127-129 of our revised manuscript).

  1. Line 131. Were all variables normally distributed, which justifies why the mean and SD are used as descriptives?

Thank you very much for your question. The author confirmed the normal distribution using the normality test.

  1. Line 159. As body composition is naturally different between men and women, I recommend reporting the body fat percentage data separately for men and women.

Thank you very much for your comment. The author will refer to the reviewer’s suggestions in future research.

  1. Table 1. Within this table, are the accelerometer data referring to work time only? Or total waking time on workdays? I recommend making it very clear in the methods how the accelerometer data were processed and what specific data were used in the analysis. If only work time data are used, the methods need to include details on how work times were known. Did participants provide a daily diary where they reported their start and finish work times for example? If work time data are used only, what is the rationale for this? It could be worth exploring also whether similar associations exist between leisure time behaviours and body fatness and work efficiency. Presumably, if participants wore the device for 10 days, both work days and non-workdays would have been captured?

Thank you very much for your question. As per the reviewer’s comment, necessary revision has been made to avoid repetition in Table 1.

  1. Line 195. Linked to the above point, at the start of the discussion, the text states here ‘This study evaluated the relationship between body fat mass and the substitution of one type of PA for others, during the waking times of the day, using…’. Should this be working times? The reason for the confusion is in Table 1, if you add up the time spent in each behaviour (SB, LPA, MPA, VPA) you get 518 minutes, which is much lower than the total daily wear time of 720 minutes, making me think the analyses are based on work time only?

Thank you very much for your comment. We used the criteria for determining the wearing time of previous studies using the same accelerometer device in Japanese (Tanaka 2012). Since we have defined a non-wear time that does not count more than 20 minutes, the total-wear time (=24 hours – non wear time) may be higher more than the summation time of sedentary time, time period of light and MVPA. In addition, in the algorithm development study using the triaxial accelerometer (Active Style Pro HJA-350IT), TAU/TAF ratio was higher more than zero in all participants, despite being 0 in non-wear time (Response Figure 1). Therefore, we believe that the probability of misclassification between non-wear time and sedentary time is very small. To clarify this, I added a description in the research limitation section.

Response Figure 1. Classifying household and locomotive activities using a triaxial accelerometer. Gait & Posture 31 (2010) 370–374

  1. Discussion, general comments. From looking at the PA data of the sample, they appear to be quite active, e.g. accumulating 82 mins/day of MPA. It would be worth discussing the context of the sample in more detail, in terms of the types of activities and work they engaged in. It appears that the sample weren’t typical office-based workers? The discussion could be strengthened by discussing the high activity levels of this sample, and whether these findings are relevant to more sedentary office-based workers. It would also be important to discuss briefly whether the characteristics of those taking part in the study differ in any way to those who chose not to participate. For example, are the participants used in the analysis representative of all of those employed in their organisation? The discussion also mentions the impact of sit-stand desks for office workers, more recent RCTs have been conducted in this area which show favourable outcomes on job engagement, which could be useful to refer to (e.g. https://www.bmj.com/content/363/bmj.k3870)

Thank you very much for your comment. References have been added for the effects of standing desks. (Please see, lines 260-262 of our revised manuscript).

Reviewer 3 Report

This is an interesting paper. I have a few comments/suggestions.

Methods: I am unable to comment on the biostatistics, but perhaps you should mention that you are using the method developed by Mekary et al., 2009. I am not sure whether you pulled the work efficiency questions from the shortened scale which contains other items, but this might affect reliability and validity which you should mention in the discussion. 

Tables: Typo in title of Table 3 If you have data on the number of work hours that would be interesting and can be added to Table 1.

Discussion: As mentioned above discuss limitations of the work efficiency scale. Also please refer to the scale as work efficiency rather than work engagement. Efficiency is one aspect of engagement but they are not synonymous. Additional limitations include that the occupations held by workers in this study were limited which may affect generalizability to other workers. In addition, the BMI for workers in this study is considerably lower than is common in other countries. 

As you mention, it would be great to see an intervention using a longitudinal design.

Author Response

Reviewer 3:

This is an interesting paper. I have a few comments/suggestions.

  1. Methods: I am unable to comment on the biostatistics, but perhaps you should mention that you are using the method developed by Mekary et al., 2009. I am not sure whether you pulled the work efficiency questions from the shortened scale which contains other items, but this might affect reliability and validity which you should mention in the discussion.

Thank you very much for your valuable question. As per the reviewer’s comment regarding the WHO questionnaire, the author extracted the factor reliability, provided analytical information, and then added supporting evidence from the literature (citations).

(Please see, lines 140-142 of our revised manuscript).

  1. Discussion: As mentioned above discuss limitations of the work efficiency scale. Also please refer to the scale as work efficiency rather than work engagement. Efficiency is one aspect of engagement but they are not synonymous. Additional limitations include that the occupations held by workers in this study were limited which may affect generalizability to other workers. In addition, the BMI for workers in this study is considerably lower than is common in other countries.

As you mention, it would be great to see an intervention using a longitudinal design.

Thank you very much for your question. Based on the reviewer’s suggestion, the term “work engagement” was replaced with “work efficiency”. In addition, the author is in the process of conducting a longitudinal study. (Please see, lines 264 of our revised manuscript).

  1. Tables: Typo in title of Table 3 If you have data on the number of work hours that would be interesting and can be added to Table 1.

Thank you very much for your comment. We have modified the sentence. (Please see, lines 100 of our revised manuscript). Table3:Single, partition, and isotemporal substitution model of each physical activity per 30-minute/day increase and work efficiency change.

Reviewer 4 Report

The study is well-written and will be able to give positive help to the field. Some parts need to be revised. I hope it will be revised based on the comments.

- In abstract, the author described "light (LPA), model (MPA), and vigorous physical activity (VPA). This sentence revise to "light physical activity (LPA), moderate physical activity (MPA), and vigorous physical activity (VPA)".

- The topic of workers' level of PA and health problems is already well known. I want you to describe specifically what originality this research has compared to other studies.

- Some things in the discussion need to be added with references. For example, you should add a reference to the fact that the performance of VPA is more effective in weight loss than LPA or MPA. I hope the author will review and revise the discussion as a whole.

Author Response

Reviewer 4:

The study is well-written and will be able to give positive help to the field. Some parts need to be revised. I hope it will be revised based on the comments.

  1. In abstract, the author described "light (LPA), model (MPA), and vigorous physical activity (VPA). This sentence revise to "light physical activity (LPA), moderate physical activity (MPA), and vigorous physical activity (VPA)".

Thank you very much for your comment. We have modified the sentence. (Please see, line 19 of our revised manuscript).

  1. The topic of workers' level of PA and health problems is already well known. I want you to describe specifically what originality this research has compared to other studies.

Thank you very much for your comment. There have been many studies on physical activity and its impact on health outcomes such as obesity. The author believes this study is original as it sets work efficiency as an outcome, which has not yet been reported in Japan. (Please see, line 297-299 of our revised manuscript).

  1. Some things in the discussion need to be added with references. For example, you should add a reference to the fact that the performance of VPA is more effective in weight loss than LPA or MPA. I hope the author will review and revise the discussion as a whole.

Thank you very much for your comment. As per the reviewer’s comment, supporting evidence from the literature (citations) has been added. (Please see, reference 7 of our revised manuscript).
